# Diffusion Coefficients and Activation Energies of Diffusion of Organic Molecules in Poly(lactic acid) Films

**DOI:** 10.3390/molecules30092064

**Published:** 2025-05-06

**Authors:** Johann Ewender, Rafael Auras, Uruchaya Sonchaeng, Frank Welle

**Affiliations:** 1Fraunhofer Institute for Process Engineering and Packaging (IVV), 85354 Freising, Germany; johann.ewender@ivv.fraunhofer.de; 2School of Packaging, Michigan State University, East Lansing, MI 48824, USA; aurasraf@msu.edu (R.A.); uruchaya.s@ku.ac.th (U.S.); 3Department of Packaging and Materials Technology, Faculty of Agro-Industry, Kasetsart University, Bangkok 10900, Thailand

**Keywords:** barrier testing, activation energy of diffusion, free volume, biodegradable

## Abstract

Poly(lactic acid) (PLA) is one of the most important bio-based and industrial compostable materials in food packaging. Its barrier properties towards oxygen and moisture are well documented. However, data on barrier properties of PLA towards organic molecules are scarce in the literature. This study investigated the diffusion of various organic molecules, including *n*-alkanes, 1-alcohols, 2-ketones, ethers, esters, amines, and aromatics, in two commercial PLA films with thicknesses of 20 µm and 30 µm. The diffusion coefficient (*D*_P_) values were determined from lag time in permeation tests conducted at temperatures ranging from 20 °C to 90 °C. The films were also characterized in terms of crystallinity, rigid and mobile amorphous fractions, and molecular weight. Activation energies (*E*_A_) were calculated based on the temperature dependence of the *D*_P_ using the Arrhenius approach. In total, 290 *D*_P_ values for 55 individual substances were determined, and 38 *E*_A_ values were derived from these data. The *E*_A_ correlated well with the molecular volume of the investigated substances. Moreover, the pre-exponential factor *D*_0_ showed a correlation with *E*_A_. These correlations enabled the establishment of diffusion modeling parameters for PLA, allowing the prediction of *D*_P_ for untested substances. The diffusion behavior of PLA was further compared with the literature data for polyethylene terephthalate and polyethylene naphthalate, providing insights into the relative performance of these materials.

## 1. Introduction

Poly(lactic acid)—PLA—is a leading bio-based and compostable plastic material widely used in food packaging and agricultural applications. Its mechanical, thermal, and structural properties, as well as its barrier performance against oxygen and moisture, have been extensively studied [1,2]. Adequate oxygen and moisture barriers are essential for extending the shelf life of packaged foods. Additionally, the diffusion of organic molecules through PLA determines its effectiveness as a barrier to flavoring substances and influences the migration of substances from PLA packaging into food. However, studies on the barrier properties of PLA towards organic molecules and diffusion coefficients of organic substances in PLA remain scarce because measuring organic vapor diffusion is challenging and requires specialized instrumentation and methodologies [3]. An overview of PLA barrier properties has been published, reporting the lack of these mass transfer data and the lack of consideration of PLA as a three-phase structure comprising a crystalline fraction, a mobile amorphous fraction, and a rigid (or restricted) amorphous fraction (RAF) [2]. The presence of RAF in semicrystalline PLA plays a crucial role in mass transfer due to the increased free volume, particularly at temperatures below the glass transition temperature, affecting the diffusion of gases and perhaps organic molecules, especially [4].

The migration of polymer ingredients, monomers, additives, and non-intentionally added substances from PLA packaging materials into food or food simulants was also investigated [5,6,7,8]. In addition, the sensory properties of PLA are relevant for the food law compliance evaluation of packed products in PLA packaging [9]. Understanding the diffusion properties of organic molecules in PLA is critical for assessing the transfer of substances from packaging materials into the products. Predictive modeling approaches have been developed to estimate the migration of monomers and additives from packaging into food using diffusion coefficients (*D_P_*). For conventional fossil-based polymers like polyolefins, polystyrene, polyethylene terephthalate, polyamide, etc., such modeling parameters have been well established [10,11]. For these polymers, diffusion modeling can be used to support conformity assessment or product development. Therefore, the migration prediction can support or even greatly simplify the assessment of packaging safety and regulatory compliance. Diffusion coefficients in bio-based polymers are also of interest in other areas such as bioengineering, e.g., in the nanostructure of bone [12], cartilage [13], or the extracellular space of the brain [14]. However, these modeling parameters have not yet been systematically determined for bio-based polymers like PLA. Diffusion modeling parameters are typically derived from migration experiments or experimentally determined *D_P_*, which can be correlated with the molecular weight or molecular volume of the substances. Such correlations enable the prediction of *D_P_* for untested molecules [10,15].

This study aimed to determine the diffusion coefficients of organic substances in PLA using permeation experiments, specifically lag time tests. The tested substances included a range of organic compounds with varying polarity and functional groups. The activation energies of diffusion (*E_A_*) for these molecules were then calculated from the temperature dependence of *D_P_*. From the *E_A_*, the diffusion modeling parameters were derived.

## 2. Results

### 2.1. Film Characterization

#### 2.1.1. Differential Scanning Calorimetry (DSC)

The tested PLA films (PLA-S and PLA-N) were characterized for their crystallinity (*X_C_*) and rigid and mobile amorphous fractions (*X_RAF_* and *X_MAF_*, respectively). Figure 1 shows representative DSC thermograms. As shown in Table 1, all thermal properties of the films, glass transition temperature (*T*_g_), cold crystallization temperature (*T_c_*), and melting temperature (*T_m_*), except for the *X_MAF_* percentage, exhibit statistical differences between PLA-S and PLA-N. Both films are predominantly amorphous.

#### 2.1.2. Fourier-Transform Infrared (FTIR) Spectroscopy

The FTIR absorbance spectra of PLA-S and PLA-N are provided in the Appendix A. Despite differences in peak intensities, both films’ peak positions remain identical, confirming their identical chemical structures. The variations in peak intensities are likely due to differences in film thickness and sample variability.

#### 2.1.3. Gel Permeation Chromatography

Gel permeation chromatography (GPC) was used to determine the molecular weight of the PLA films. Table 2 presents the number average molecular weight (*M_n_*), weight average molecular weight (*M_w_*), and dispersity (*Đ_M_*) of PLA-S and PLA-N. The *Đ_M_* values were calculated as *M_w_*/*M_n_*. The *M_n_* values of PLA-S and PLA-N differ significantly, while their *M_w_* values do not.

### 2.2. Diffusion Coefficient Determination

The diffusion coefficients (*D_P_*) of organic substances like *n*-alkanes, 1-alcohols, 2-ketones, ethers, esters, amines, and aromatics were determined from the lag time through the two commercial biaxially oriented PLA films, PLA-S and PLA-N. For the 20 μm PLA film (PLA-S), 57 individual *D_P_* were determined. Furthermore, 233 *D_P_* were measured for the 30 µm film (PLA-N). Overall, 290 *D_P_* for 55 organic substances were determined within this study for both investigated PLA films. The *D_P_* were assessed between 20 °C and 90 °C. However, due to the low diffusivity of organic substances in PLA, most of the *D_P_* were measured between 70 °C and 90 °C, above the *T*_g_ of PLA, as indicated in Table 3. Only the gaseous alkanes (methane, ethane, and *n*-propane) could be measured at temperatures below the *T*_g_ of PLA in a reasonable time. For higher molecular weight substances, the permeation rates for the tested organic compounds were too low and, therefore, below the analytical detection limits of the applied lag time method. The experimentally determined diffusion coefficients are given in the Appendix A. *D_P_* values for the same compounds, such as 1-octanol, 1-butanol, and *n*-hexane for both samples, are comparable and between experimental errors.

The activation energies of diffusion (*E*_A_) were derived from the temperature dependence of *D_P_*. *E_A_* was only calculated if the following criteria were met to determine the activation energies as accurately as possible: at least four diffusion coefficients were available, showing good linearity in the Arrhenius plot (r^2^ > 0.95) over at least a 15 °C temperature range. If the temperature range is too small, the slope of the linear correlation in the Arrhenius plot is too inaccurate, and the *E_A_* calculated thereof is consequently also inaccurate. The *E_A_*, as well as the pre-exponential factors, *D*_0_, are presented in Table 3 for PLA-S and Table 4 for PLA-N.

## 3. Discussion

The experimentally determined *D_P_* values for the 55 individual molecules of different polarity, functional groups, as well as molecular size, can be considered as representative of typical permeants of migrants of PLA. In general, large molecules diffuse slower through a polymer matrix compared with smaller molecules of a similar structure. As expected, higher molecular volume *V* results in lower *D_P_* at a given temperature. More-over, higher temperature results in higher *D_P_*, which is expected from an Arrhenius relationship. However, due to the low diffusivity of PLA, the largest molecule tested in this study was di-*iso*-propyl naphthalene, with a molecular weight of 212.3 g/mol (molecular volume 227.94 Å^3^). Larger molecules therefore diffuse very slowly in the polymer so that the lag time is very long. For example, the lag time of di-*iso*-propyl naphthalene at 80 °C and 20 µm film is about 3 days. At the lower temperature of 60 °C, the lag time of di-*iso*-propyl naphthalene increases to approximately 18 years, as predicted for the diffusion modeling approach established in this study. We can assume a thinner PLA film of 10 µm will decrease the lag time to about 4.5 years, but this is still too long for an experimental determination of the lag time. In addition, thin films of 10 µm are not commercially available. The experimental measurement window is therefore very limited and is a condition determined by molecule size, film thickness, and temperature. Furthermore, due to the low diffusivity of PLA, almost all *D_P_* in this study were measured above the glass transition temperature, where the PLA is in the rubbery state. This means that the diffusion is higher than in the glassy state and is an additional overestimating factor for predicting the migration below *T_g_*.

As a result, the experimentally determined *D_P_* show a very strong dependence on the temperature. The *E_A_* were calculated from this dependence. The *E_A_* and the *D*_0_ values for each permeant were calculated from the Arrhenius relationship below or above *T*_g_. The results show good linearity for the investigated permeants, indicating that the diffusion process follows Fick’s laws of diffusion. Figure 2 shows the correlations between *E_A_* and the molecular volume *V*, as well as between *E_A_* and the pre-exponential factor *D*_0_, respectively. The experimental data for PLA are shown in comparison with the published activation energy data for polyester PET [15] and PEN [16]. For all three polyester polymers, the *E_A_* values are in the same order of magnitude for small molecules up to *V* of approximately 100 Å^3^ and for *E*_A_ of about 120 kJ/mol. For larger molecules, the *E*_A_ values for PLA are significantly higher than those of PET and PEN. The *D*_0_ values for all three polyesters follow a strong linear relationship over the range of approximately 20 orders of magnitude for PET and PEN, and approximately 40 orders of magnitude for PLA. The slope of the *E_A_* and *D*_0_ correlation for PLA is different from the slopes determined for PET and PEN. Compared to PET and PEN, the increase in activation energy with molecule size is significantly more substantial for PLA. This means that even small molecules exhibit high activation energies of diffusion. A high activation energy implies that the temperature has a much stronger influence on diffusion. Therefore, increasing temperature has a high influence on the migration of substances into food or food simulants. This increase is not as significant for PET and PEN. However, like PET and PEN, PLA is a low-diffusible polymer which only releases substances into the food in small quantities unless high-swelling food simulants are used.

The two correlations can be used to determine the diffusion modeling parameters for PLA for Equation (1). This equation was developed for PET [15] but also applies to PLA, and the correlations can be established for PLA as well (Figure 2). According to Equation (1), the *D_P_* can be predicted from the molecular volume *V* with the diffusion modeling parameters *a* to *d* for PLA. The parameters *a* to *d* are the slope and the axis intercept, respectively, in the two correlations for PLA. The parameter *a* is the slope of the correlation between pre-exponential factors *D*_0_ and *E_A_*, with *b* being the intercept of this correlation. Parameters *c* and *d* are the intercept and slope of the correlation between *V* and *E_A_*. The diffusion modeling parameters derived from the two correlations are given in Table 5 along with the previously reported values for PET [15] and PEN [16]. In Equation (1), *V* is the molecular volume and *T* is the temperature in Kelvin.(1)$$D_{P}=b{\left(\frac{V}{c}\right)}^{\frac{a-\frac{1}{T}}{d}}$$

Figure 3 shows a comparison of the experimentally determined *D_P_* with the predicted *D_P_* from Equation (1) and the parameters *a* to *d* from Table 5. As a result, the slope of the correlation is near 1 and the straight line almost passes through the origin, which indicates the *D_P_* were predicted realistically and well over the molecular size and temperature range. However, there is a certain spread in the data of experimentally determined *D_P_* versus predicted *D_P_*, possibly due to slight differences among the PLA-S and PLA-N films. The two dashed lines represent the 95% confidence interval of the correlation. Previously, it has been shown that not only the crystallinity of the films—in this case, both mostly amorphous—but also the *X_RAF_*, albeit small compared with *X_MAF_*, and *M_n_* play a role in *D_P_* [17,18].

A comparison of the experimentally determined diffusion coefficients with data from the literature [2] is very difficult. On the one hand, there is very little data available. Secondly, the permeation rates were mostly determined while the diffusion coefficients were not. However, the few *D_P_* measured previously are generally higher than the *D_P_* measured in this study. It is very likely the permeation is measured at higher concentrations, which leads to interactions between the permeant and the polymer in the case of PLA, which probably increases the measured *D_P_*.

## 4. Materials and Methods

### 4.1. Materials

Two commercial biaxially oriented PLA films were used in this study:PLA-S: A 20-µm biaxially oriented PLA film (Evlon^®^, BI-AX International Inc., Wingham, ON, Canada).PLA-N: A 30-µm biaxially oriented PLA film (Nativia^®^, Taghleef Industries, Newark, DE, USA).

### 4.2. Film Characterization

#### 4.2.1. Differential Scanning Calorimetry (DSC)

A differential scanning calorimeter, Q100 (TA Instruments, New Castle, DE, USA), calibrated with indium standards, was used for characterization and investigation of the crystallinity of the films. The film samples, 5–10 mg, were weighed and sealed in an aluminum pan, and the thermal analyses were performed in nitrogen atmosphere with a flow rate of 70 mL/min. For film characterization, the samples were cooled from room temperature to −50 °C, heated from −50 to 200 °C (first heating cycle), maintained isothermally for 1 min at 200 °C, cooled until −50 °C, and finally heated to 200 °C (second heating cycle). The temperature ramp rate for all the cycles was 10 °C/min. The characterization for each film was run in triplicate and the results were analyzed with the TA Instruments Universal Analysis 2000 software version 4.5A to determine the *T*_g_, *T_c_*, and *T_m_*, as well as the enthalpies associated with these temperatures based on results from the second heating cycle. The degree of crystallinity (*X_C_*) was calculated from Equation (2):(2)$$X_{C}=\frac{{\Delta H}_{m}-{\Delta H}_{c}}{{\Delta H}_{f}^{0}}$$
where Δ*H_m_* and Δ*H_c_* are the enthalpy of melting and the enthalpy of cold crystallization of the sample, respectively. Δ*H_f_*^0^ is the enthalpy of fusion of pure crystalline sample, which is 93.1 J/g for PLA [19]. The mobile amorphous fraction (*X_MAF_*) was calculated from Equation (3):(3)$$X_{MAF}=\frac{{\Delta C}_{p}}{{\Delta C}_{p}^{0}}$$
where Δ*C_p_* and Δ*C_p_*^0^ are the changes in heat capacity at *T_g_* for the semicrystalline sample and the fully amorphous sample, respectively. Δ*C_p_*^0^ for a fully amorphous PLA is 0.639 J/(g °C) [20]. The rigid amorphous fraction (*X_RAF_*) was calculated from Equation (4):(4)$$X_{RAF}=1-{X_{C}-X}_{MAF}$$

#### 4.2.2. Fourier-Transform Infrared Spectroscopy

Chemical functional groups in the films were identified using Fourier-transform infrared (FTIR) spectroscopy (IRAffinity-1S, Shimadzu, Columbia, MD, USA) in both transmittance and attenuated total reflectance (ATR) modes. Each spectrum was recorded with a resolution of 2 per cm and a mirror speed of 2.8 mm/s, with a total of 20 scans per measurement.

#### 4.2.3. Gel Permeation Chromatography

Molecular weight distribution was determined by gel permeation chromatography (GPC) using a Waters 1515 system (Waters, Milford, MA, USA) equipped with a refractive index detector (Waters 2414) and HR Stryragel^®^ columns of 7.8 mm × 300 mm (HR4, HR3, and HR2). Approximately 10 mg of PLA was dissolved in 5 mL of tetrahydrofuran (THF) (Pharmco-Aaper, Brookfield, CT, USA), filtered through a 0.45 µm PTFE filter (Simsii, Inc., Port Irvine, CA, USA), and injected into a 2 mL GPC glass vial. All GPC runs were eluted with THF at a flow rate of 1.0 mL/min at 35 °C for 50 min and an injection volume of 100 µL. A calibration curve was constructed using polystyrene standards with the Mark–Houwink constants K = 0.0164 mL/g and α = 0.704. Data were analyzed using Waters Breeze2^®^ software version 3.30. Each sample was analyzed in triplicate.

#### 4.2.4. Statistical Analysis

Statistical analyses of DSC and GPC data were performed by one-way analysis of variance (ANOVA), followed by Tukey’s multiple comparison test at a 95% confidence level (α = 0.05) using SAS^®^ Studio Release 3.8 (SAS Institute Inc., Cary, NC, USA).

### 4.3. Diffusion Coefficient Determination

#### 4.3.1. Permeants

Two types of permeants were tested: liquid permeants and permanent gases.

Liquid permeants: *n*-alkanes (*n*-pentane to *n*-tetradecane), 1-alcohols (1-propanol to 1-octanol), 2-ketones (acetone to 2-octanone), cyclic alkanes, ethers, ketones (tetrahydrofuran, cyclohexane, 4-vinylcyclohexene, 1,3-dioxolane, 2-methyl-1,3-dioxolane, 1,4-dioxane, cyclopentane, cyclopentanone, cyclohexanone), esters (formic acid methyl ester to formic acid heptyl ester), amines (pyridine, pyrrol, 2,6-lutidine, aniline, 2-aminobenzonitrile), and aromatic substances (benzene, toluene, ethylbenzene, *n*-propylbenzene, *n*-butylbenzene, naphthalene, 1-methylnaphthalene, 1-ethylnaphthalene, 2,7-di-*iso*-propylnaphthalene, phenanthrene, anthracene). The respective groups of substances were mixed and measured together.Permanent gases: Methane, ethane, *n*-propane, and *n*-butane were tested as a mixture of gaseous alkanes.

#### 4.3.2. Permeation Cell

PLA films were clamped in a permeation steel cell between two sealant rings. The surface area of the tested films was 191 cm^2^. The permeation cell, placed in a climate chamber, was divided into a lower and an upper space by the film. The lower space had a volume of 7667 cm^3^. The permeants were introduced into the cell as follows:Liquid permeants: The liquid mixtures were injected into the lower space of the permeation cell through a septum using a syringe. Upon injection, the permeants evaporated immediately at high temperatures (65 °C to 90 °C).Permanent gases: The gaseous alkane mixture was flushed through the lower space of the permeation cell with a constant gas flow.

The upper space of the permeation cell was permanently rinsed with a pure stream of nitrogen (20 mL/min) which moved the permeated substances out of the cell. The nitrogen stream went through a connected enrichment unit and the permeants were trapped on this unit over a period of 20 min. The enrichment unit was connected to a gas chromatograph (GC) with flame ionization detection. The permeants were directly desorbed into the GC, where their quantities were determined through quantitative analysis. During the GC run, the next sample was trapped on the enrichment unit and subsequently injected into the GC. By using this method, one kinetic point was measured every 45 min.

Gas Chromatographic conditions for the different permeants were as follows:Liquid permeants: Column: Rxi 624, length: 60 m, internal diameter: 0.32 mm, film thickness: 1.8 µm, carrier gas: 120 kPa helium. Temperature program: 40 °C (2 min), rate 10 °C/min to 200 °C, rate 20 °C/min to 270 °C, and held for 8 min. Pre-trap: substances collected on 20 mm length by 5 mm diameter of Carbopack B, desorbed at 350 °C. Main trap: substances focused at −46 °C on 30 mm length by 1.4 mm diameter of Carbopack B, desorbed at 320 °C.Permanent gases: Column: Poraplot Q, length: 30 m, internal diameter: 0.53 mm, film thickness: 20 µm, carrier gas: 120 kPa helium. Temperature program: 70 °C (0.5 min), rate 30 °C/min to 150 °C, and held for 5 min. Pre-trap: substances collected on 20 mm length by 5 mm diameter of Carbopack B, desorbed at 350 °C. Main trap: substances focused at −46 °C on 30 mm length by 1.4 mm diameter of active charcoal 140–280 µm, desorbed at 320 °C.

#### 4.3.3. Diffusion Coefficient Calculation

Isostatic and quasi-isostatic methods are often used to determine the permeation of organic substances through films [21]. In these methods, a film is clamped in a permeation cell and brought into contact with a known concentration of the permeant on one side. The permeated quantity is purged with an inert carrier gas and brought for quantification, if necessary after concentration, to the detector. The lag time is the intersection of the *x*-axis of the tangent to the steady-state concentrations. The lag time can be used to calculate the *D_P_* according to Equation (5) [2]. Calibration was performed with injections of the applied permeants at known amounts. The diffusion coefficients *D_P_* (in cm^2^/s) were calculated from the lag times (in s) according to Equation (5), where *l* is the film thickness (in cm). The activation energies of diffusion *E_A_* were calculated from the *D_P_* at various temperatures according to the Arrhenius equation (Equation (6)) below or above *T*_g_, where *D*_0_ is the pre-exponential factor, *R* is the gas constant, and *T* is the temperature in Kelvin.(5)$$lag\;time=\frac{l^{2}}{6D_{P}}$$(6)$$D_{P}=D_{0}\;e^{-\frac{E_{A}}{RT}}$$

#### 4.3.4. Molecular Volume Calculations

The molecular volumes were calculated using the freely accessible program Molinspiration [22]. The program calculates the molecular volumes using a group-based contribution method.

## 5. Conclusions

In this study, a total of 290 diffusion coefficients were experimentally determined from the lag times of the various substances during permeation through PLA. From this, 38 activation energies of diffusion were calculated. The diffusion coefficients *D_P_* as well as the activation energies of diffusion *E_A_* correlate well with the molecular volume *V* of the permeants. The pre-exponential factors *D*_0_ also correlate with the *E_A_*. If other molecules are following such correlations, the *D_P_* and *E_A_* of other substances in PLA can be predicted from the *V*. The data are used to establish the diffusion modeling parameters for PLA. These modeling parameters can now be used to predict both the permeation of organic substances through PLA and the migration of monomers, additives, or other substances from PLA. The *D_P_* determined in this study are actual diffusion coefficients. However, in migration studies, which are usually carried out in food simulants, other effects such as swelling of the packaging surface may occur, which increase diffusion and thus migration [8,23,24]. In real foods, however, such swelling effects occur less frequently.

Furthermore, the prediction of migration using diffusion modeling should always be overestimated [11,25]. The overestimation factor in *D_P_* predicted from Equation (1) and the modeling parameters in Table 5 are 1.3 on average (median 1.0). The *D_P_* are therefore predicted very well from the diffusion parameters *a* to *d* and Equation (1). This is shown in the log–log plot (Figure 3), which shows a slope near 1 and where the correlation line runs almost completely through the origin. At the same time, however, this means that almost half of the *D_P_* underestimate the migration or permeation. To achieve a slight overestimation, the molecule size can be virtually reduced to predict higher diffusion coefficients. Based on the experimental data, a 30% molecular size reduction appears sufficient as a worst-case scenario. This reduction of the molecular volume by 30% results in a mean overestimation factor of 9.2 (median 4.6). In practice, this appears to be a good overestimation factor for predicting the *D_P_* with a sufficient overestimation of the actual migration. This means the overestimation factor in migration is about 3, because the diffusion coefficient correlates with the square root of the migration.

## Figures and Tables

**Figure 1 molecules-30-02064-f001:**
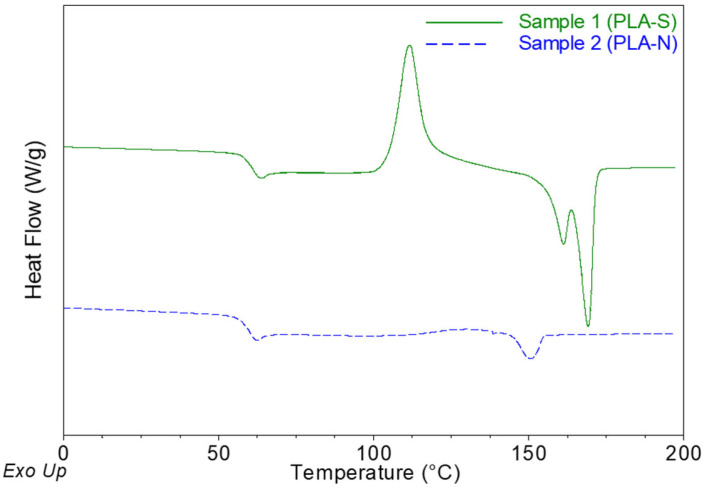
Thermograms of sample 1 (PLA-S) and sample 2 (PLA-N).

**Figure 2 molecules-30-02064-f002:**
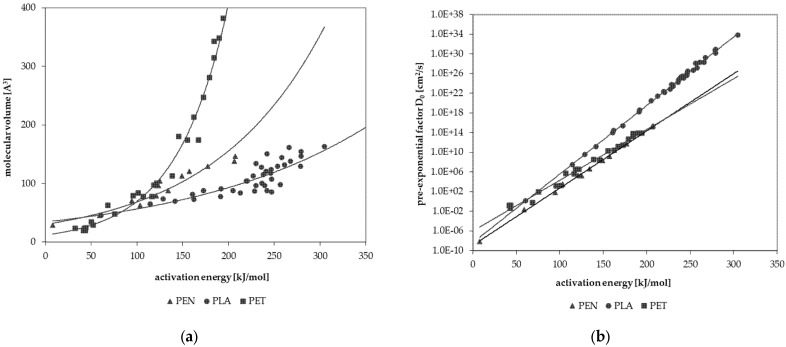
Correlations between the activation energy of diffusion *E_A_* and the molecular volume *V* of the test permeants (**a**), and correlations between *E_A_* and the pre-exponential factor, *D*_0_ (**b**), for PLA (this study), PET [15], and PEN [16].

**Figure 3 molecules-30-02064-f003:**
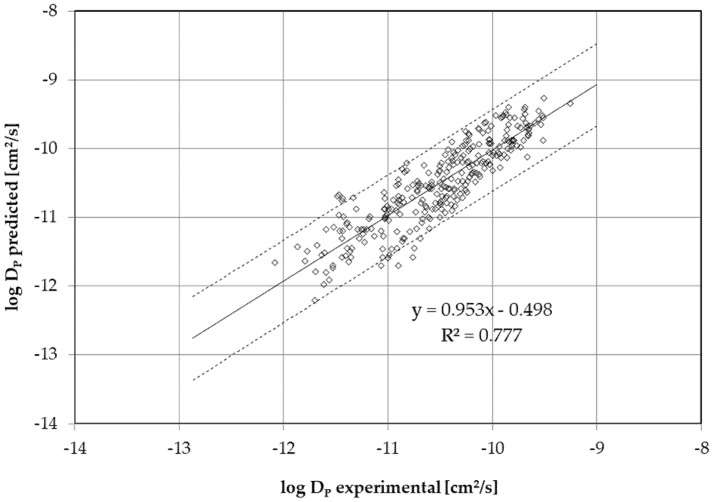
Log–log plot of the experimental and predicted diffusion coefficients for PLA. Dashed lines represent the 95% confidence interval.

**Table 1 molecules-30-02064-t001:** Thermal properties of the investigated PLA film samples.

Sample	Thickness, µm	*T_g_*, °C	*T_c_*, °C	*T_m_*_1_, °C	*T_m_*_2_, °C	*X_C_*, %	*X_MAF_*, %	*X_RAF_*, %
PLA-S	20.5 ± 0.6 ^a^	61.3 ± 0.5 ^a^	110.4 ± 1.4 ^a^	160.9 ± 0.4	169.1 ± 0.1 ^a^	3.4 ± 1.1 ^a^	88.3 ± 2.2 ^a^	8.3 ± 2.6 ^a^
PLA-N	30.0 ± 0.9 ^b^	60.1 ± 0.2 ^b^	128.9 ± 1.0 ^b^	n.d.	151.1 ± 0.3 ^b^	1.2 ± 0.2 ^b^	86.6 ± 1.9 ^a^	12.1 ± 0.8 ^b^

Note: n.d. = not detected. Values in the same column with the same lowercase letter are not significantly different based on Tukey’s HSD test at a 5% significance level.

**Table 2 molecules-30-02064-t002:** Number average molecular weight (*M_n_*), weight average molecular weight (*M_w_*), and dispersity (*Đ_M_*) of PLA film samples measured by GPC.

Sample	*M_n_*, kDa	*M_w_*, kDa	*Đ_M_*
PLA-S	84.5 ± 1.0 ^a^	175.3 ± 0.7 ^a^	2.1 ± 0.02 ^a^
PLA-N	104.4 ± 2.9 ^b^	171.5 ± 3.9 ^a^	1.6 ± 0.04 ^b^

Note: Values in the same column with the same lowercase letters are not significantly different at α = 0.05.

**Table 3 molecules-30-02064-t003:** Results for the activation energies of diffusion and the pre-exponential factor for PLA-S films.

Substance	Molecular Weight, g/mol	Molecular Volume, Å^3^	Temperature Range Tested, °C	Number of *D_P_*	Activation Energy *E_A_*, kJ/mol	Pre-Exponential Factor *D*_0_, cm^2^/s
1-propanol	60.1	70.82	70–75	3		
*n*-pentane	72.2	96.16	70–85	4	230.0	4.11 × 10^23^
1-butanol	74.1	87.62	70–85	5	172.6	2.38 × 10^15^
*n*-hexane	86.2	112.96	75–90	4	226.8	6.88 × 10^22^
1-pentanol	88.2	104.42	70–85	5	220.4	1.35 × 10^22^
*n*-heptane	100.2	129.77	75–90	4	253.9	4.32 × 10^26^
1-hexanol	102.2	121.22	75–85	3		
*n*-octane	114.2	146.57	75–90	4	279.3	1.56 × 10^30^
1-heptanol	116.2	138.03	75–85	3		
*n*-nonane	128.3	163.37	75–90	4	304.7	6.41 × 10^33^
1-octanol	130.2	154.83	75–85	3		
*n*-decane	142.3	180.17	80–90	3		
*n*-undecane	156.3	196.97	80–90	3		
*n*-dodecane	170.3	213.78	80–90	3		
*n*-tridecane	184.4	230.58	80–90	3		
*n*-tetradecane	198.4	247.38	80–90	3		

**Table 4 molecules-30-02064-t004:** Results for the activation energies of diffusion and the pre-exponential factor for PLA-N films.

Substance	Molecular Weight, g/mol	Molecular Volume, Å^3^	Temperature Range Tested, °C	Number of *D_P_*	Activation Energy *E_A_*, kJ/mol	Pre-Exponential Factor *D*_0_, cm^2^/s
methane	16.0	28.64	20	1		
ethane	30.1	45.76	20–60	4	60.6	1.47 × 10^0^
*n*-propane	44.1	62.56	50–70	3		
acetone	58.1	64.74	60–75	5	114.5	3.02 × 10^7^
*n*-butane	58.1	79.36	70	1		
methyl formate	60.1	57.16	60–70	3		
1-propanol	60.1	70.82	65–80	4	162.0	2.81 × 10^14^
pyrrole	67.1	69.03	70	1		
cyclopentane	70.1	85.80	70–85	4	247.0	1.41 × 10^26^
tetrahydrofuran	72.1	77.98	70–85	4	191.3	1.73 × 10^18^
2-butanone	72.1	81.54	60–85	10	160.7	9.09 × 10^13^
*n*-pentane	72.2	96.16	70–85	9	239.4	1.92 × 10^25^
1,3-dioxolane	74.1	70.17	60–85	6	141.4	1.47 × 10^11^
ethyl formate	74.1	73.97	60–75	4	128.7	3.87 × 10^9^
1-butanol	74.1	87.62	65–85	5	205.0	2.96 × 10^20^
benzene	78.1	84.04	70–85	4	212.8	2.45 × 10^21^
pyridine	79.1	79.89	70	1		
cyclopentanone	84.1	87.98	70–85	4	242.2	2.93 × 10^25^
cyclohexane	84.2	102.60	75–85	3		
2-pentanone	86.1	98.34	65–80	5	256.6	1.10 × 10^28^
*n*-hexane	86.2	112.96	75–85	8		
2-methyl-1,3-dioxolane	88.1	86.75	65–80	4	228.4	5.77 × 10^23^
1,4-dioxane	88.1	86.97	70–85	3		
*n*-propyl formate	88.1	90.77	60–80	5	191.9	3.60 × 10^18^
1-pentanol	88.2	104.42	70–85	4	219.3	2.43 × 10^22^
toluene	92.1	100.61	70–85	3	236.6	4.81 × 10^24^
aniline	93.1	95.33	70	1		
cyclohexanone	98.1	104.79	75–85	3		
2-hexanone	100.2	115.15	70–85	5	238.2	1.02 × 10^25^
*n*-heptane	100.2	129.77	75–90	9	278.9	5.12 × 10^30^
*n*-butyl formate	102.1	107.57	65–80	4	247.1	3.29 × 10^26^
1-hexanol	102.2	121.22	70–85	4	241.0	2.53 × 10^25^
ethylbenzene	106.2	117.41	70–90	5	246.0	6.78 × 10^25^
4-vinyl cyclohexene	108.2	124.17	75–90	4	246.4	4.31 × 10^25^
2-heptanone	114.2	131.95	70–85	5	261.4	2.05 × 10^28^
*n*-octane	114.2	146.57	80–90	6		
*n*-pentyl formate	116.2	124.37	70–80	3		
1-heptanol	116.2	138.03	70–85	4	267.5	1.66 × 10^29^
*n*-propylbenzene	120.2	134.21	75–90	4	230.2	2.44 × 10^23^
naphthalene	128.2	128.03	75–90	4	236.3	1.56 × 10^24^
2-octanone	128.2	148.75	75–85	4		
*n*-nonane	128.3	163.37	80–90	6		
*n*-hexyl formate	130.2	141.17	70–80	3		
1-octanol	130.2	154.83	70–85	4	279.4	7.62 × 10^30^
*n*-butylbenzene	134.2	151.01	75–90	4	242.3	1.21 × 10^25^
methyl naphthalene	142.2	144.60	75–90	4	258.0	1.35 × 10^27^
*n*-decane	142.3	180.17	80–90	6		
*n*-heptyl formate	144.2	157.97	75–80	2		
ethyl naphthalene	156.2	161.40	75–90	4	266.2	1.75 × 10^28^
*n*-undecane	156.3	196.97	80–90	6		
*n*-dodecane	170.3	213.78	80–90	6		
*n*-tridecane	184.4	230.58	80–90	6		
*n*-tetradecane	198.4	247.38	80–90	6		
di-iso-propyl naphthalene	212.3	227.94	80–90	3		

**Table 5 molecules-30-02064-t005:** Diffusion modeling parameters for PLA from Equation (1) compared to published values for PET and PEN.

Parameter	PLA	PET ^a^	PEN ^b^
*a* [1/K]	2.67 × 10^−3^	1.93 × 10^−3^	2.23 × 10^−3^
*b* [cm^2^/s]	4.46 × 10^−9^	2.37 × 10^−6^	1.12 × 10^−9^
*c* [Å^3^]	34.4	11.1	31.1
*d* [1/K]	4.14 × 10^−5^	1.50 × 10^−4^	6.74 × 10^−5^

^a^ Values reported in [15]; ^b^ values reported in [16].

## Data Availability

Data are contained within the article and Appendix A.

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
