# Peer review of "Diffusion Coefficients and Activation Energies of Diffusion of Organic Molecules in Poly(lactic acid) Films"

_molecules, 2025, doi:10.3390/molecules30092064_

Round 1

Reviewer 1 Report

Comments and Suggestions for Authors

This article provides rich information on the barrier properties of PLA toward organic molecules and the analysis of the experimental data is also very thorough. This is an interesting study. However, the following corrections are required before publication.

  1. Page 1 Line 27: In my opinion, the keyword of compostable is not suitable for this article. Although PLA is one of the compostable material, most of this article emphasized on the barrier properties of PLA as the food packaging material.

  1. It seems like two melting peaks appear in the curve of PLA-S in Figure 1, while there is only one Tm value (169.1±0.1 ℃) in Table 1. Please check and conform it.

  1. How are the precise volumes of these organic molecules in Tables 3 and Table 4 obtained?

  1. The description about lag time including it definition, its determination method and units, etc., should be supplied in the methods part 4.3, especially in the text in front of Equation 5.

  1. Page 6 Line 166: “The parameters a to d are the slope and the axis intercept, respectively, in the two correlations for PLA”, this expression of the parameters of a, b, c, and d in Equation 1 was not clear and unambiguous.

  1. The multiplication signs are missing in Table 5.

  1. It is suggested that the part of “The DP determined in this study are actual diffusion coefficients.…… PLA is a low-diffusible polymer which only releases substances into the food in small quantities unless highly swelling food simulants are used.” in conclusion section moved in the discussion section to condense and simplify the expression of conclusion.

  1. There are only dashed lines but not solid lines in the Figure S7. Please check the annotation of Figure 7. Actually, Figure S7 should be Figure S6.

Author Response

reviewer 1

This article provides rich information on the barrier properties of PLA toward organic molecules and the analysis of the experimental data is also very thorough. This is an interesting study. However, the following corrections are required before publication.

    Page 1 Line 27: In my opinion, the keyword of compostable is not suitable for this article. Although PLA is one of the compostable material, most of this article emphasized on the barrier properties of PLA as the food packaging material.

Response of the authors: We removed the keyword compostable.

    It seems like two melting peaks appear in the curve of PLA-S in Figure 1, while there is only one Tm value (169.1±0.1 ℃) in Table 1. Please check and conform it.

Response of the authors: The DSC results in Table 1 (Page 3, Lines 79–81) have been revised to include both Tm₁ and Tm₂ for PLA-S, in accordance with the two melting peaks observed in Figure 1.

    How are the precise volumes of these organic molecules in Tables 3 and Table 4 obtained?

Response of the authors: The molecular volumes were calculated with the free program molinspiration. We added an additional chapter in Materials and Methods.

    The description about lag time including it definition, its determination method and units, etc., should be supplied in the methods part 4.3, especially in the text in front of Equation 5.

Response of the authors: We have explained the procedure for determining the diffusion coefficients a little better.

    Page 6 Line 166: “The parameters a to d are the slope and the axis intercept, respectively, in the two correlations for PLA”, this expression of the parameters of a, b, c, and d in Equation 1 was not clear and unambiguous.

Response of the authors: We have explained the parameters a to d better in the text

    The multiplication signs are missing in Table 5.

Response of the authors: We added the multiplication signs in Table 5

    It is suggested that the part of “The DP determined in this study are actual diffusion coefficients.…… PLA is a low-diffusible polymer which only releases substances into the food in small quantities unless highly swelling food simulants are used.” in conclusion section moved in the discussion section to condense and simplify the expression of conclusion.

Response of the authors: We have moved this part from the Conclusions to the Discussion section.

    There are only dashed lines but not solid lines in the Figure S7. Please check the annotation of Figure 7. Actually, Figure S7 should be Figure S6.

Response of the authors: We updated Figure S6 (formerly S7)

Reviewer 2 Report

Comments and Suggestions for Authors

The study addresses a knowledge gap by presenting extensive experimental data on diffusion coefficients and activation energies of organic molecules in PLA, contributing to the understanding of mass transfer processes in PLA packaging. The methodology is well articulated, with detailed experimental setups and robust analytical techniques. The authors provide a substantial dataset (290 Diffusion coefficient values for 55 substances), offering valuable parameters for modelling transport processes in PLA.

The analysis performed in the study aligns with the journal’s focus on materials sciences making it highly suitable for publication.

However, several suggestions could further enhance the manuscript.

The introduction effectively outlines the gap regarding organic molecule diffusion in PLA. However, a brief comparison with fossil-based polymers is mentioned without sufficient detail. Expanding this comparison slightly could enhance the reader’s context.

Moreover, including examples of diffusion analysis in various research fields could broaden the relevance of the paper. Studies on diffusion in complex heterogeneous media are also conducted in bioengineering, e.g. in bone nanostructure [1], cartilage [2] or brain extracellular space [3]. Consider citing the work of Bini et al. [1], that presents relevant methodologies for modeling diffusion phenomena at nanoscale within structured polymeric and biological matrices. Including this work would underscore complementary computational strategies and enhance the broader scientific context.

References:

[1] Bini, F. et al. 2021. 3D random walk model of diffusion in human Hypo- and Hyper- mineralized collagen fibrils. Journal of Biomechanics, Volume 125, 110586. doi: 10.1016/j.jbiomech.2021.110586

[2] Momot, K.I.. 2011. Diffusion tensor of water in articular cartilage. Eur. Biophys. J. 40, 81–91. doi: 10.1007/s00249-010-0629-4.

[3] Jin, S.,et al. 2008. Random-Walk model of diffusion in three dimensions in brain extracellular space: comparison with microfiberoptic photobleaching measurements. Biophys. J. 95, 1785–1794. doi: 10.1529/biophysj.108.131466.

In the discussion section, it would be beneficial for the authors to explicitly compare their findings with those from previous studies regarding diffusion in PLA to contextualize their results better.

I also recommend that the authors better highlight the limitations of their study.

Author Response

reviewer 2

The study addresses a knowledge gap by presenting extensive experimental data on diffusion coefficients and activation energies of organic molecules in PLA, contributing to the understanding of mass transfer processes in PLA packaging. The methodology is well articulated, with detailed experimental setups and robust analytical techniques. The authors provide a substantial dataset (290 Diffusion coefficient values for 55 substances), offering valuable parameters for modelling transport processes in PLA.

The analysis performed in the study aligns with the journal’s focus on materials sciences making it highly suitable for publication.

However, several suggestions could further enhance the manuscript.

The introduction effectively outlines the gap regarding organic molecule diffusion in PLA. However, a brief comparison with fossil-based polymers is mentioned without sufficient detail. Expanding this comparison slightly could enhance the reader’s context.

Response of the authors: We have given examples of fossil-based polymers here and described the advantage of modelling parameters for bio-based polymers a little better.

Moreover, including examples of diffusion analysis in various research fields could broaden the relevance of the paper. Studies on diffusion in complex heterogeneous media are also conducted in bioengineering, e.g. in bone nanostructure [1], cartilage [2] or brain extracellular space [3]. Consider citing the work of Bini et al. [1], that presents relevant methodologies for modeling diffusion phenomena at nanoscale within structured polymeric and biological matrices. Including this work would underscore complementary computational strategies and enhance the broader scientific context.

Response of the authors: Thank you very much for this tip. We have included this link to other subject areas in the introduction and cited the three papers.

References:

[1] Bini, F. et al. 2021. 3D random walk model of diffusion in human Hypo- and Hyper- mineralized collagen fibrils. Journal of Biomechanics, Volume 125, 110586. doi: 10.1016/j.jbiomech.2021.110586

[2] Momot, K.I.. 2011. Diffusion tensor of water in articular cartilage. Eur. Biophys. J. 40, 81–91. doi: 10.1007/s00249-010-0629-4.

[3] Jin, S.,et al. 2008. Random-Walk model of diffusion in three dimensions in brain extracellular space: comparison with microfiberoptic photobleaching measurements. Biophys. J. 95, 1785–1794. doi: 10.1529/biophysj.108.131466.

In the discussion section, it would be beneficial for the authors to explicitly compare their findings with those from previous studies regarding diffusion in PLA to contextualize their results better.

Response of the authors: We have added a discussion of the diffusion coefficients

I also recommend that the authors better highlight the limitations of their study.

Response of the authors: This is a very generalised point that is difficult to implement in the manuscript without more specific information. We moved parts of the conclusions to the discussion section to emphasise the limitations (e.g. that swelling can influence the diffusion coefficients and cause them to deviate from the predicted ones) more prominently.